# Risk Factors for Mortality in Adult COVID-19 Patients Who Develop Bloodstream Infections Mostly Caused by Antimicrobial-Resistant Organisms: Analysis at a Large Teaching Hospital in Italy

**DOI:** 10.3390/jcm10081752

**Published:** 2021-04-17

**Authors:** Brunella Posteraro, Giulia De Angelis, Giulia Menchinelli, Tiziana D’Inzeo, Barbara Fiori, Flavio De Maio, Venere Cortazzo, Maurizio Sanguinetti, Teresa Spanu

**Affiliations:** 1Dipartimento di Scienze Biotecnologiche di Base, Cliniche Intensivologiche e Perioperatorie, Università Cattolica del Sacro Cuore, 00168 Roma, Italy; brunella.posteraro@unicatt.it (B.P.); giulia.deangelis78@gmail.com (G.D.A.); giulia.menchinelli@hotmail.it (G.M.); tiziana.dinzeo@unicatt.it (T.D.); cortazzo.venere@gmail.com (V.C.); teresa.spanu@unicatt.it (T.S.); 2Dipartimento di Scienze Mediche e Chirurgiche, Fondazione Policlinico Universitario A. Gemelli IRCCS, 00168 Roma, Italy; 3Dipartimento di Scienze di Laboratorio e Infettivologiche, Fondazione Policlinico Universitario A. Gemelli IRCCS, 00168 Roma, Italy; barbara.fiori@policlinicogemelli.it (B.F.); demaioflavio@yahoo.it (F.D.M.)

**Keywords:** COVID-19, bloodstream infection, mortality, risk factors, antimicrobial resistance, septic shock

## Abstract

The aim of this study was to characterize COVID-19 (SARS-CoV-2-infected) patients who develop bloodstream infection (BSI) and to assess risk factors associated with in-hospital mortality. We conducted a retrospective observational study of adult patients admitted for ≥48 h to a large Central Italy hospital for COVID-19 (1 March to 31 May 2020) who had or had not survived at discharge. We included only patients having blood cultures drawn or other inclusion criteria satisfied. Kaplan–Meier survival or Cox regression analyses were performed of 293 COVID-19 patients studied, 46 patients (15.7%) had a hospital-acquired clinically relevant BSI secondary to SARS-CoV-2 infection, accounting for 58 episodes (49 monomicrobial and 9 polymicrobial) in total. Twelve episodes (20.7%) occurred at day 3 of hospital admission. Sixty-nine species were isolated, including *Staphylococcus aureus* (32.8%), Enterobacterales (20.7%), *Enterococcus faecalis* (17.2%), *Candida* (13.8%) and *Pseudomonas aeruginosa* (10.3%). Of 69 isolates, 27 (39.1%) were multidrug-resistant organisms. Twelve (54.5%) of 22 patients for whom empirical antimicrobial therapy was inappropriate were infected by a multidrug-resistant organism. Of 46 patients, 26 (56.5%) survived and 20 (43.5%) died. Exploring variables for association with in-hospital mortality identified > 75-year age (HR 2.97, 95% CI 1.15–7.68, *p* = 0.02), septic shock (HR 6.55, 95% CI 2.36–18.23, *p* < 0.001) and BSI onset ≤ 3 days (HR 4.68, 95% CI 1.40–15.63, *p* = 0.01) as risk factors independently associated with death. In our hospital, mortality among COVID-19 patients with BSI was high. While continued vigilance against these infections is essential, identification of risk factors for mortality may help to reduce fatal outcomes in patients with COVID-19.

## 1. Introduction

While facing an unprecedented viral pandemic with over 2.1 million deaths globally as of 27 January 2021 [1], clinicians are now dealing with bacterial or fungal coinfections/superinfections [2,3,4,5], including antimicrobial-resistant infections [6], that complicate and/or aggravate the course of coronavirus disease 2019 (COVID-19) [7]. Unlike co-infection—defined as the simultaneous infection with a microbial pathogen other than severe acute respiratory syndrome coronavirus 2 (SARS-CoV-2)—secondary infection or superinfection occurs in patients during or following SARS-CoV-2 infection [8].

Usually, superinfections are hospital-acquired (or nosocomial) infections caused by bacterial or fungal (e.g., *Candida* species) pathogens often displaying a multidrug-resistant phenotype (e.g., carbapenemase-producing Enterobacterales) [9]. Importantly, these infections have the potential to increase mortality among hospitalized patients with COVID-19 [10,11,12], probably because the interaction between SARS-CoV-2 and superinfecting (or secondary) microbial organism(s) may increase SARS-CoV-2-induced tissue destruction and/or facilitate systemic dissemination of co-pathogens [13]. It is thus likely that prescribed antimicrobial drugs in COVID-19 patients, to reduce severe or fatal outcomes of disease including septic shock, are often unsuccessful [14].

In the present study, we performed a retrospective assessment of clinical or microbiological factors associated with mortality in adult COVID-19 patients who developed bloodstream infection (BSI) in a large teaching hospital in Central Italy. In particular, we explored the impact of BSI caused by an antimicrobial-resistant bacterial or *Candida* organism.

## 2. Materials and Methods

### Study Design, Data Collection and Definitions

We included adult (aged ≥18 years) patients admitted to the Fondazione Policlinico Universitario A. Gemelli (FPG) IRCCS tertiary care hospital in Rome (Italy) in the period between 1 March and 31 May 2020 if they had a laboratory-confirmed COVID-19 diagnosis (i.e., tested positive for SARS-CoV-2 via nasopharyngeal swab RT-PCR assay) [15] and had blood cultures drawn on or after admission (Figure 1). We initially excluded those patients who had (i) no blood cultures drawn, (ii) no clinical data available or (iii) a hospital stay <24 h. Patients with blood cultures positive for not clinically relevant microbial species were definitely excluded (Figure 1). In this retrospective observational study of COVID-19 patients, we mainly focused on 46 patients with BSI; however, we also performed an ad hoc comparison between these patients and 50 (randomly selected) patients without BSI.

After collection, blood cultures were processed for up to five days—after that, negative cultures for microbial growth were discarded—or until positivity (assessed within five days) using the BacT/ALERT VIRTUO system (bioMérieux, Marcy l’Étoile, France). For BSI isolates, identification was performed using the MALDI Biotyper^®^ system (Bruker Daltonics, Bremen, Germany), and antimicrobial susceptibility testing (AST) was performed using VITEK^®^ 2 (bioMérieux, Marcy-l’Étoile, France) or, only for *Candida* isolates, Sensititre™ YeastOne™ (Thermo Fisher Scientific, Waltham, MA, USA) systems. EUCAST clinical breakpoints [16] were used to interpret minimum inhibitory concentration (MIC) values to all routinely tested antimicrobial drugs, except for those of fluconazole or echinocandin antifungal drugs that were interpreted according to clinical breakpoints reported in the CLSI M27-S4 document [17]. Antibiotic resistance phenotypes were confirmed by retesting bacterial isolates with the EUCAST broth microdilution method [18], whereas antimicrobial resistance genes/determinants were characterized for both bacterial and *Candida* isolates using PCR sequencing or other previously described methods [19,20]. We defined multidrug-resistant isolates as those isolates that displayed non-susceptibility to at least one drug in three or more antimicrobial classes.

We used medical records from the study patients to retrieve (and collect) pertinent clinical information, which included patient age, gender and comorbidities at the time of admission, laboratory values (e.g., inflammatory markers), sequential organ failure assessment (SOFA) score, septic shock [21], COVID-19 status (i.e., moderate, severe or critical) [22], blood culture results, empirical antimicrobial therapies, admission to ICU, length of ICU or hospital stay and in-hospital mortality. All collected data were de-identified and handled to ensure anonymity and confidentiality. Two senior microbiologists (B.P. and T.S.) assessed the clinical relevance of blood culture results. The assessment relied on either clinical information available from medical records or the likelihood of skin contamination based on the growth of coagulase-negative staphylococci (e.g., *Staphylococcus epidermidis*) or other common skin colonizers (e.g., *Corynebacterium*) from only one blood culture drawn from a unique sampling site (e.g., culture from central venous catheter blood only) [23]. Antibiotic or antifungal drugs were defined as both empirical, if received within the first five days after blood culture collection (which is the time interval a blood culture was detected as positive for microbial growth), and appropriate, if compared with AST results of bacterial or fungal isolates. BSI onset was categorized as hospital-acquired or ICU-acquired if a positive blood culture was recorded on or after the third day of hospital or ICU admission, respectively [24]. BSI was also categorized as polymicrobial when a blood culture grew more than one organism or a different organism within 48 h of the first positive blood culture [25]. Positive blood cultures from the same patient were considered subsequent episodes of BSI if occurred 14 or more days after the first blood culture positive for the same organism. Sources of bacteremia or fungemia were identified from infectious disease specialist notes and confirmed independently by two microbiologists (G.D.A. and T.S.).

With respect to hospital admission for COVID-19, diagnoses of BSI at the time of or within the first 24 h were defined as coinfections, and those that occurred ≥48 h after were categorized as superinfections [23]. For patients who had been transferred from other hospitals or healthcare facilities, the first day of admission to the FPG hospital was at least equivalent to the third day of stay in the FPG hospital.

## 3. Data Analysis

Data were analyzed using Stata software version 13.0 (StataCorp, College Station, TX, USA). Continuous data were presented as mean (standard deviation (SD)) or median (interquartile range (IQR)) and categorical data as counts and percentages. Differences between a priori established groups according to survival status (i.e., survivors versus non-survivors) were assessed using a *t*-test or Mann–Whitney *U*-test (as appropriate) for continuous variables or using Chi-square test or Fisher’s exact test (as appropriate) for categorical variables. Survival was measured from the date of hospital admission until all-cause death (event) or hospital discharge (censored), and the Kaplan–Meier method with the log-rank test was used to assess the effect of variables on survival. Variables with a *p* < 0.05 in the univariable Cox regression analysis, as well as variables with known prognostic value, were assessed in a multivariable Cox regression analysis with a stepwise forward selection method. Independent predictors of survival were modeled using a Cox proportional-hazard analysis, and the Schoenfeld residuals plots’ inspection and the Grønnesby and Borgan’s test were used to assess the proportional-hazard assumption or the model goodness of fit, respectively. Statistical significance was measured by a two-sided *p* value of <0.05.

## 4. Results

As shown in Figure 1, 293 patients with confirmed COVID-19 had blood cultures drawn during their hospital stay. Of these patients, 215 (73.4%) had negative blood cultures. Of 78 patients with positive blood cultures, 46 (46/293; 15.7%) were infected by clinically relevant microbial species. In 32 (32/293; 10.9%) remaining patients, blood cultures grew microbial species deemed to be contaminants and were then excluded from the analysis. In particular, 53 (24.7%) of 215 patients without and 37 (80.4%) of 46 patients with positive blood cultures were sampled >48 h after hospital admission. Nine of 46 patients were admitted to our hospital within 48 h since discharge from another hospital or healthcare facility where COVID-19 was diagnosed. Thus, with respect to COVID-19 diagnosis, all 46 patients had a BSI secondary to SARS-CoV-2 infection. Of 46 patients, 26 (56.5%) survived and 20 (43.5%) died.

### 4.1. Characteristics of COVID-19 Patients Who Developed BSI

Table 1 shows the characteristics of 46 patients with BSI stratified by whether patients were discharged alive from (*n* = 26) or had died in (*n* = 20) the hospital.

There was no significant difference between survivors and non-survivors in age, gender or presence of Charlson or other comorbidities. Comparison of infection-related characteristics showed that non-survivors had a higher level of procalcitonin (*p* = 0.02) and were more likely to have septic shock (*p* < 0.001) or earlier BSI onset (*p* = 0.02) than survivors. Comparison of clinical outcomes showed that, expectedly, survivors stayed in the ICU (*p* = 0.03) or in the hospital (*p* < 0.001) longer than non-survivors.

Ten of 46 patients had multiple BSI infection episodes (see Appendix A). As detailed in Figure 2, among 58 episodes studied in total, 12 (20.7%) occurred at day 3 (range, 1–3 days), six (10.4%) at day 10 (5–9 days), 13 (22.4%) at day 20 (11–19 days), 13 (22.4%) at day 30 (22–30 days) or 14 (24.1%) at day >30 (33–110 days) of hospital admission. 

### 4.2. Characteristics of BSI Episodes in COVID-19 Patients

A respiratory source was identified in 20 (34.5%) of 58 episodes. The most frequently isolated species were, in decreasing order, *Staphylococcus aureus* (19/58; 32.8%), Enterobacterales (12/58; 20.7%), *Enterococcus faecalis* (10/58; 17.2%), *Candida* species (8/58; 13.8%) and *Pseudomonas aeruginosa* (6/58; 10.3%). Additionally, Enterobacterales (4/9; 44.4%), *Enterococcus faecalis* (4/9; 44.4%) or *Candida* species (3/9; 33.3%) were most frequently isolated from polymicrobial episodes.

Overall, 27 (39.1%) of 69 BSI isolates were multidrug-resistant organisms (see Appendix A). Fifteen (78.9%) of 19 *S. aureus* isolates were oxacillin-resistant (methicillin-resistant *S. aureus* (MRSA)) whereas one (50.0%) of two *E. faecium* isolates was vancomycin-resistant. Two (16.7%) of 12 Enterobacterales isolates were third-generation cephalosporin-resistant (one *Escherichia coli* and one *Klebsiella pneumoniae*). Carbapenem resistance was found in three (25.0%) of 12 Enterobacterales isolates and in two (100%) of two *Acinetobacter baumannii* isolates whereas three (50.0%) of six *P. aeruginosa* isolates were resistant to antipseudomonal cephalosporins or carbapenems. One (11.1%) of nine *Candida* isolates was echinocandin-resistant (one *Candida glabrata*). Molecular analyses revealed the presence of *mecA* in MRSA isolates, *bla*_CTX-M-15_ in third-generation cephalosporin-resistant Enterobacterales isolates, *bla*_KPC_ in carbapenem-resistant Enterobacterales isolates, *vanA* in the vancomycin-resistant *E. faecium* isolate, *bla*_OXA-23_ in carbapenem-resistant *A. baumannii* isolates and a mutated *FKS* in the echinocandin-resistant *C. glabrata* isolate.

Regarding empirically administered antimicrobial drugs (see Appendix A), 15 (32.6%) of 46 patients with BSI did not receive antimicrobial drugs similarly to 20 (40.0%) of 50 patients without BSI—randomly selected from those included in the study—at the time of first blood culture collection. Patients were most frequently treated with piperacillin/tazobactam (alone or together with linezolid or vancomycin) (14/46; 30.4% versus 9/50; 18.0%) or carbapenems (alone or together with linezolid or vancomycin) (13/46; 28.3% versus 2/50; 4.0%). Conversely, 19 (38.0%) of 50 patients without BSI, as opposed to none (0.0%) of 46 patients with BSI, received ceftriaxone (alone in four patients or together with azithromycin in 15 patients). Among seven (15.2%) of 46 patients inappropriately treated, four patients (infected by *E. faecium*, *bla*_KPC_ positive *K. pneumoniae*, MRSA or carbapenem-resistant *P. aeruginosa*, respectively), two patients (infected by *bla*_CTX-M-15_ positive *E. coli* or MRSA, respectively) and one patient (infected by *bla*_OXA-23_ positive *A. baumannii*) received carbapenems, piperacillin/tazobactam or piperacillin/tazobactam together with linezolid, respectively. Twelve (54.5%) of 22 patients with inappropriate empirical antimicrobial therapy (15 untreated and 7 treated) and 10 (41.7%) of 24 patients with appropriate empirical antimicrobial therapy were infected by a multidrug-resistant organism (see Appendix A).

### 4.3. Risk Factors Associated with In-Hospital Mortality among BSI Patients

Appendix A summarizes the results of Kaplan–Meier or Cox regression analyses for 46 COVID-19 patients with BSI. 

Survival percentages among patients who had >75-year age, C-reactive protein > 90 mg/L, septic shock or BSI onset ≤ 3 days were significantly lower than among patients who had ≤75-year age (*p* = 0.02), C-reactive protein ≤ 90 mg/L (*p* = 0.02), no septic shock (*p* < 0.001) or BSI onset >3 days (*p* = 0.001). Multivariable logistic regression identified > 75-year age (hazard ratio (HR) 2.97, 95% confidence interval (CI) 1.15–7.68, *p* = 0.02), septic shock (HR 6.55, 95% CI 2.36–18.23, *p* < 0.001) and BSI onset ≤ 3 days (HR 4.68, 95% CI 1.40–15.63, *p* = 0.01) as risk factors independently associated with death. As shown in Figure 3, Kaplan–Meier curves of the 46 patients stratified by the presence or absence of septic shock or by the ≤3 or >3-day time to BSI onset showed that survival rates for patients with septic shock or with ≤3-day time to BSI onset were 46.0% and 36.0%, respectively, at day 21 of hospital stay. Considering the overall COVID-19 patient cohort (*n* = 622) during the study period (see Figure 1), survival rate among 576 patients without BSI (84.0%) was higher than among 46 patients with BSI (71.0%) at day 21 of hospital stay (*p* = 0.03).

## 5. Discussion

Our retrospective single-center analysis of adult patients who developed a positive blood culture-based clinically relevant BSI during hospitalization for COVID-19 found a 43.5% (20/46) rate of in-hospital mortality. This rate was significantly higher than that of COVID-19 patients without positive blood cultures (24.2%; 52/215) who were hospitalized during the study period (1 March to 31 May 2020) (not shown; Chi-square test, *p* = 0.008). Excluding 32 patients with positive blood cultures for likely contaminants, 189 (26/46 and 163/215) of 293 (64.5%) eligible patients survived during the study period. Our assessment of risk factors for in-hospital mortality among the 46 patients with BSI found >75-year age, septic shock or ≤3-day time to BSI onset as the only independent factor associated with death.

Unlike previous studies [23,26,27], we did not restrict our analysis to the first 48 h of hospital admission and did not include infections other than BSI as secondary infections (also named superinfections) in COVID-19 patients. We specifically focused on BSI because this infection with its clinical consequences (i.e., sepsis or septic shock) [28] remains a major cause of mortality among hospitalized patients [29]. Therefore, when presenting our experience of superinfections in patients with COVID-19, we provided a careful description of 46 patients with their 58 BSI episodes, including antimicrobial susceptibility or resistance profiles of 69 BSI causative isolates. In our study, 78/293 COVID-19 patients (26.6%) for whom blood culture was performed had a bacterial/fungal infection, of which 32/78 (41.0%) were categorized as contaminants. In the study by Hughes et al. [30] on 836 COVID-19 patients admitted to two London (UK) hospitals, 60/643 (9.3%) for whom blood culture was performed had a bacterial infection, of which 39/60 (65.0%) were categorized as contaminants. Similarly, in the study by Garcia-Vidal et al. [23] on 989 COVID-19 patients admitted to one Barcelona (Spain) hospital, 16/267 (6.0%) for whom blood culture was performed had a bacterial infection. In that study [23], coagulase-negative staphylococci caused 43.7% of BSIs (7/16) whereas *S. aureus* was only isolated from patients with hospital-acquired pneumonia, and two other superinfections were BSI due to *Candida albicans*. In our study, *S. aureus* caused 32.8% of BSIs (19/58) with 78.9% of isolates (15/19) being MRSA, and eight other BSIs were due to *Candida* species with 12.5% of isolates (1/8) being an echinocandin-resistant *C. glabrata*. Of note, the last infection occurred in a patient with previous BSI episodes, of which the first was due to MRSA and the second due to multidrug-resistant Gram-negative bacteria [20]. Antimicrobial-resistant organisms caused a high rate of BSIs in our study. These findings place emphasis on local stewardship strategies to reduce the rates of empirical antimicrobial therapy in suspected SARS-CoV-2 patients as well as to denote the need for antipseudomonal and/or atypical therapy. The knowledge of the spread of antimicrobial resistance within the hospital setting calls for sustained surveillance systems, which remain an important healthcare focus to limit unintended consequences of the COVID-19 pandemic [31].

With respect to secondary *S. aureus* bacteremia, our findings mirror those reported by Cusumano et al. [10] who described 42 cases in COVID-19 patients admitted to two New York City (USA) hospitals during the same three months (March 1 to May 31, 2020) as in our study. Similar to us, and unlike previous reports on *S. aureus* bacteremia [32,33], Cusumano et al. [10] studied the timing and other relevant features of bacteremia (including pneumonia source) or the association with in-hospital mortality. Importantly, that case series found 14-day and 30-day hospital mortality rates of 54.8% (23/42) and 66.7% (28/42) and, at multivariable analysis, hospital-onset bacteremia and age as significant predictors of 14-day mortality and Pitt bacteremia score as a significant predictor of 30-day mortality [10]. Unlike in the Cusumano et al. study [10], variables associated with lower survival rates in our case series—including inappropriate empirical antimicrobial therapy and antimicrobial-resistant infection—were explored for association with in-hospital mortality. Of variables showing significantly increased hazards of death (i.e., older age, higher C-reactive protein level, septic shock or earlier onset of BSI) at univariable analysis, septic shock and earlier onset of BSI—both assessed as infection-related characteristics—remained significantly associated with in-hospital mortality at multivariable analysis (Appendix A).

Sepsis or septic shock are common complications in COVID-19 patients, especially those requiring ICU admission [12,34,35,36], and might directly result from SARS-CoV-2 infection [37]. In a risk assessment for death in adult COVID-19 patients from Wuhan (China) hospitals [36], sepsis preceded development of secondary bacterial infection in 27 (96.4%) of 28 patients who did not survive, whereas septic shock occurred only in patients who did not survive (38/54 (70.4%) versus 0/137 (0.0%); *p* < 0.0001). Consistently, only one of 26 patients discharged alive from our hospital had septic shock compared to 12 (60.0%) of 20 patients who did not survive (*p* < 0.001; Table 1). In particular, analysis of survival by subgroups (Figure 3) showed that only 46% of patients with septic shock were still alive 21 days after hospital admission. This emphasizes the importance to limit the progression of bacterial secondary infection in SARS-CoV-2 infected patients [13]. In that study, Zhou et al. [36] assessed septic shock independently on secondary infection and found that a higher SOFA score—which is a proxy for sepsis and septic shock [21]—was associated with increased odds of in-hospital death. Conversely, we assessed septic shock in our 46 patients at the time of their documented sepsis (i.e., with an identified microbial organism), and it is likely that SARS-CoV-2 could have been a contributor to secondary bacterial sepsis (and not just a single causative agent of sepsis), as already observed with influenza virus and staphylococcal sepsis [37]. However, pathogenesis of sepsis in COVID-19 remains unclear.

It is also plausible that virus-induced direct tissue or cell damage in the lower respiratory tract of our COVID-19 patients—all with a moderate to critical pneumonia—could have favored an early (herein defined as ≤3-day from hospital admission) BSI onset in some rather than in other patients. Unknown reasons might have caused those patients to be more disposed to the entry of microbial pathogens into the bloodstream, probably because of an altered mucosal barrier function that in turn may be caused by patient-specific immunological or clinical factors [38]. In particular, patients with sepsis or septic shock may develop a leaky gut that enables translocation of bacterial or *Candida* organisms (and/or their components) from the gut to systemic circulation [5]. Whether this increased disposition may result in increased risk for disease severity and mortality in certain patients with COVID-19 remains not understood. At present, current evidence suggests that SARS-CoV-2 may not significantly affect bacterial or fungal virulence [39]. Although our risk factors analysis did not reveal any association with mortality for inappropriate empirical antimicrobial therapy, it is worth noting that 15 of 46 patients did not receive any antimicrobial drug at the time of first blood collection and seven of 46 patients received inappropriate therapy. Reasons for these findings are understandable. On one side, uncertainty to distinguish COVID-19 and its progression from bacterial or fungal infection might have limited clinicians in prescribing antimicrobials [39], which should indeed be recommended in mild to moderate COVID-19 cases without clear indication of co-infection/superinfection [40]. On the other side, inappropriateness was unavoidable in four patients treated with carbapenems and in three patients treated with piperacillin/tazobactam because of the intrinsic or acquired resistance to respective antimicrobials of BSI causative agents (Appendix A).

We acknowledge that our study has some major limitations. First, this was a study with a retrospective design and small sample size, which limit data collection and control over multiple confounders. We only included clinically relevant BSI episodes microbiologically documented, and it is possible that either some episodes have been missed or some in-charge physicians did not have ordered blood cultures for their patients. Therefore, the patient selection process for inclusion in the study may be biased. Additionally, we did not perform a strict variable selection for multivariable Cox regression analysis to consider the total number of deaths in our study and to avoid overfitting in the Cox regression model. Second, this study was limited to a single hospital, which may restrict the generalizability of the presented findings. To mitigate this issue, we specifically discussed our findings within the context of a case series from patients hospitalized for COVID-19 in the same temporal period as in our study. Finally, consistent with the recently published literature, we reported on the first wave of the COVID-19 pandemic. Consequently, empirical use of antimicrobial agents and ensuing frequency and etiology of BSIs might be no longer reflective of those at the time of writing, when the COVID-19 pandemic is raging.

In conclusion, our study suggests that mortality among hospitalized patients with COVID-19 who develop BSI secondary to SARS-CoV-2 infection is high, as well as is the rate of BSIs due to antimicrobial-resistant organisms eventually selected by initial (empirical) antimicrobial therapies. Risk factors like those identified in the present study may help clinicians to identify at an early stage patients with COVID-19 who have poor prognosis following secondary BSIs. While vigilance against bacterial or fungal infections is essential in the management and treatment of COVID-19 patients, rapid characterization of BSI episodes is likely to decrease fatal outcomes and, meanwhile, improve antimicrobial stewardship during the COVID-19 pandemic.

## Figures and Tables

**Figure 1 jcm-10-01752-f001:**
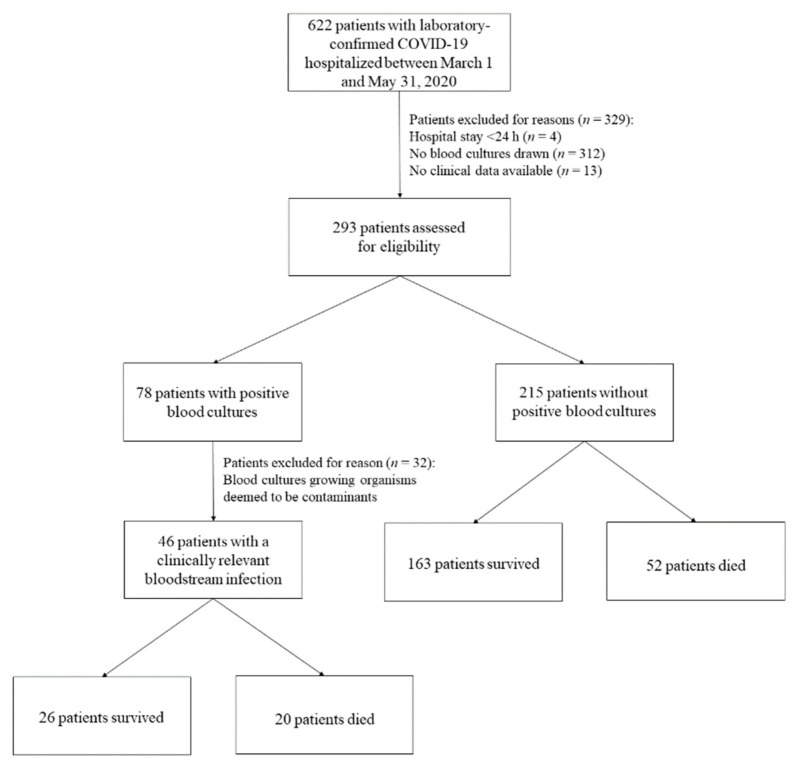
Flow chart of patient selection for inclusion in the study. Among 293 eligible patients (78 with and 215 without positive blood cultures), 46 patients were identified as having a clinically relevant bloodstream infection (BSI) and were then included.

**Figure 2 jcm-10-01752-f002:**
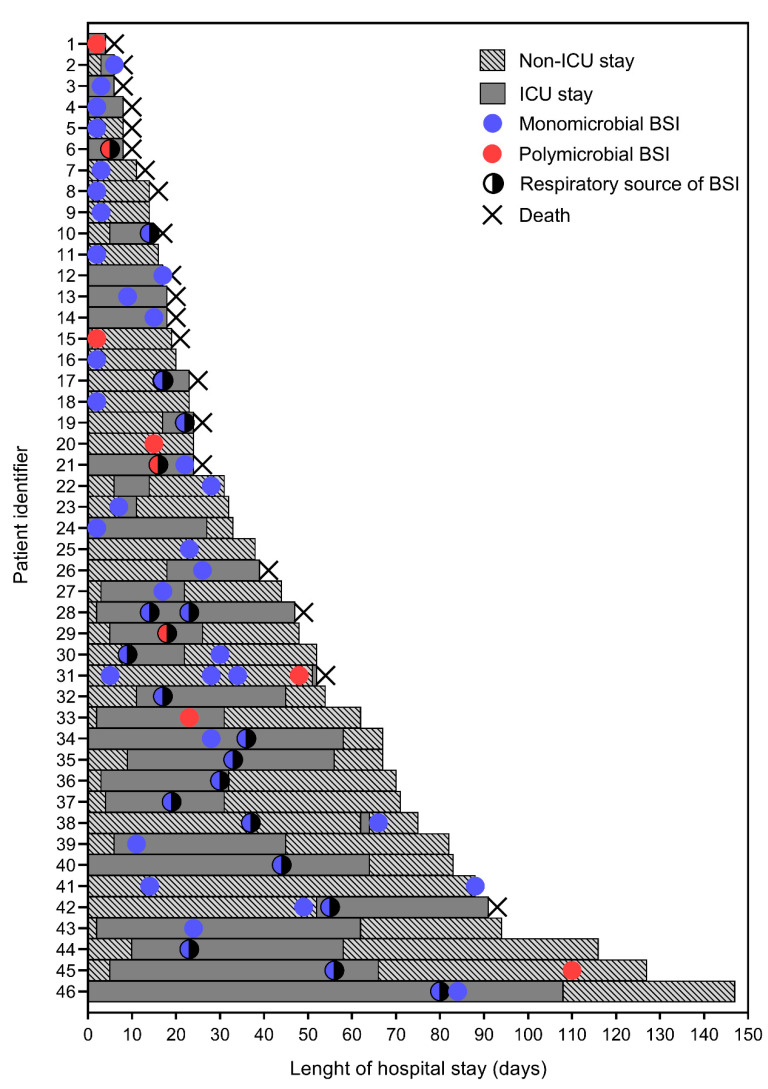
Timeline of 58 bloodstream infection (BSI) episodes relative to the length of hospital stay for 46 patients included in the study. Details about ICU (solid bars) or non-ICU (dashed bars) stays as well as the type (blue- or red-colored circle) or known respiratory source (semi-colored dark circle) of each BSI episode are shown. Multiplication sign indicates the event (i.e., death) that interrupted the length of hospital stay.

**Figure 3 jcm-10-01752-f003:**
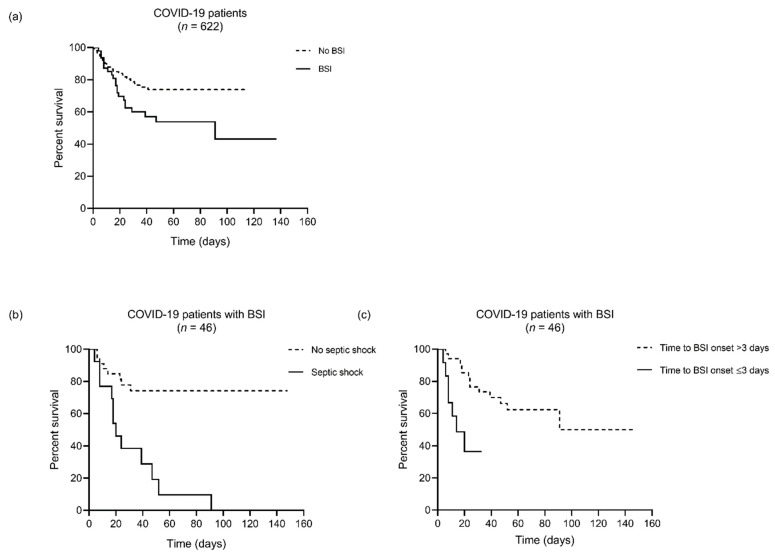
Survival of COVID-19 patients hospitalized during the study period. In (**a**), 622 patients were stratified by the presence (*n* = 46) or absence (*n* = 576) of clinically relevant bloodstream infection (BSI). In (**b**) and (**c**), 46 patients with BSI were further stratified by the presence (*n* = 13) or the absence (*n* = 33) of septic shock or by the ≤3-day (*n* = 12) or >3-day (*n* = 34) time to BSI onset, respectively.

**Table 1 jcm-10-01752-t001:** Demographic and clinical characteristics of patients with COVID-19 who developed bloodstream infection during hospital stay.

No. (%) of Patients with or Values of Characteristics by Groups				
Characteristics	All (*n* = 46)	Survivors (*n* = 26)	Non-Survivors (*n* = 20)	*p* Value
Age, years, median (IQR)	69.5 (62.0–82.0)	75.5 (64.0–83.0)	69.0 (61.0–67.0)	0.11
Male	33 (71.7)	20 (76.9)	13 (65.0)	0.37
Charlson comorbidities at admission				
Myocardial infarction	7 (15.2)	3 (11.5)	4 (20.0)	0.35
Congestive heart failure	4 (8.7)	2 (7.7)	2 (10.0)	0.59
Peripheral vascular disease	8 (17.4)	5 (19.2)	3 (15.0)	0.51
Cerebrovascular disease	3 (6.5)	1 (3.9)	2 (10.0)	0.40
Dementia	11 (23.9)	7 (26.9)	4 (20.0)	0.42
Chronic pulmonary disease	10 (21.7)	5 (19.2)	5 (25.0)	0.45
Rheumatologic disease	3 (6.5)	1 (3.9)	2 (10.0)	0.40
Diabetes without complications	7 (15.2)	3 (11.5)	4 (20.0)	0.35
Diabetes with chronic complications	4 (8.7)	2 (7.7)	2 (10.0)	0.59
Hemiplegia or paraplegia	3 (6.5)	1 (3.9)	2 (10.0)	0.40
Kidney disease	6 (13.0)	2 (7.7)	4 (20.0)	0.22
Any malignancy ^a^	6 (13.0)	4 (15.4)	2 (10.0)	0.46
Charlson comorbidity index score, mean (SD)	2.0 (1.8)	1.8 (1.5)	2.4 (2.0)	0.30
Hypertension	23 (50.0)	13 (50.0)	10 (50.0)	0.62
COVID-19 severity status ^b^				
Moderate	12 (26.1)	8 (30.8)	4 (20.0)	0.31
Severe	26 (56.5)	15 (57.7)	11 (55.0)	0.54
Critical	8 (17.4)	3 (11.5)	5 (25.0)	0.21
Infection-related characteristics				
SOFA score, median (IQR) ^c^	6.5 (3.0–9.0)	6.0 (2.0–9.0)	8.0 (4.0–9.0)	0.20
C-reactive protein, mg/L, median (IQR) ^c^	116.1 (59.6–182.9)	95.2 (59.0–163.7)	147.7 (100.3–212.7)	0.11
Procalcitonin, ng/mL, median (IQR) ^c^	0.4 (0.2–1.6)	0.2 (0.2–0.8)	0.6 (0.3–8.0)	0.02
Interleukin 6, pg/mL, median (IQR) ^c^	133.0 (40.0–487.2)	108.9 (32.2–442.8)	216.1 (81.1–652.0)	0.32
Septic shock	13 (28.3)	1 (3.9)	12 (60.0)	<0.001
Respiratory source of infection	19 (41.3)	12 (46.2)	7 (35.0)	0.32
Recurrent infection	5 (10.9)	4 (15.4)	1 (5.0)	0.26
Antimicrobial-resistant infection	24 (52.2)	13 (50.0)	11 (55.0)	0.77
ICU-acquired infection	26 (56.5)	14 (53.9)	12 (60.0)	0.45
Time to infection onset, days, median (IQR) ^c^	15.2 (2.7–23.4)	18.1 (9.4–27.7)	8.5 (1.8–17.3)	0.02
Inappropriate empirical antimicrobial therapy ^c,d^	22 (47.8)	14 (53.9)	8 (40.0)	0.26
ICU admission	36 (78.3)	19 (73.1)	17 (85.0)	0.27
ICU stay, days, median (IQR)	9 (1–27)	20 (0–39)	8 (2–18)	0.03
Hospital stay, days, median (IQR)	33 (18–70)	58 (32–82)	18 (8–28)	<0.001

COVID-19, coronavirus disease 2019; IQR, interquartile range; SD, standard deviation; SOFA, sequential organ failure assessment. ^a^ Includes solid cancer, leukemia or lymphoma. ^b^ According to the National Institutes of Health definitions [22]. ^c^ Assessed at the time of first bloodstream infection episode. ^d^ Assessed in treated (*n* = 7) or untreated (*n* = 15) patients.

## Data Availability

The data presented in this study are available on request from the corresponding author.

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
