# Peer review of "Risk Factors for Mortality in Adult COVID-19 Patients Who Develop Bloodstream Infections Mostly Caused by Antimicrobial-Resistant Organisms: Analysis at a Large Teaching Hospital in Italy"

_jcm, 2021, doi:10.3390/jcm10081752_

Round 1
Reviewer 1 Report
Congratulations to the authors on their paper which is methodologically sound and well writen. I submit some observations for your consideration.
-Consider defining Multidrug resistant in the Methods section.
-In table 3 lenght of ICU stay should probably be left out of the analysis since it is closely related to mortality.
-In table 3 and the text there is a variable called 'Time to infection onset >3 or <= 3 days', but earlier in the text it is stated that all patients had healthcare related BSI (lines 139-144) since patients with BSI >48h had been recently discharged (if I understand correctly). If so, all patients should be regarded as having nosocomial infection and this variable should be reconsidered or discussed.
-A simple statistical analisis is missing in the sentence of lines 244-246/ fig 3 (a), and lines 248-250.
-Line 347 refers to a second wave but this may be restricted to the author’s country, since other countries have experienced additional waves. Consider using another term.
-Consider addressing the high rate of MDR infections in the discussion and comment on the relationship with local susceptibility patters which may relate to empirical treatments and may not apply to other enviroments.
Author Response
Dear Sir/Madam,
I send you the revised version of the manuscript (jcm-1158044) “Risk factors for mortality in adult COVID-19 patients who develop bloodstream infections mostly caused by antimicrobial-resistant organisms: analysis at a large teaching hospital in Italy” by Posteraro et al. according to the reviewers' comments. All changes made in the manuscript were clearly highlighted, using the “Track Changes” function in Microsoft Word, so that they are easily visible to the editor and reviewers.
Below, you can find a short cover letter detailing any changes, for the benefit of the editor and reviewers. We detailed the revisions that have been made, citing the line number and exact change, so that the editor can check the changes expeditiously.
Reviewer 1
Congratulations to the authors on their paper, which is methodologically sound and well written. I submit some observations for your consideration.
-Consider defining Multidrug resistant in the Methods section.
Answer: We added the definition regarding “multidrug-resistant isolates” (i.e. “those isolates that displayed non-susceptibility to at least one drug in three or more antimicrobial classes”). See page 3, lines 107 to 109 of the revised manuscript.
-In table 3, length of ICU stay should probably be left out of the analysis since it is closely related to mortality.
Answer: As suggested, we omitted the length of ICU stay from the analysis. Accordingly, we re-performed the analysis and the resulting data (when changed) were presented in the revised Table 3 (now moved to Supplementary Material and, then, renamed Table S4; see comment 6 of the reviewer 2). See page 1, lines 28 to 30; page 8, lines 247 to 250, and Table S4 of the revised manuscript.
-In table 3 and the text there is a variable called 'Time to infection onset >3 or <=3 days', but earlier in the text it is stated that all patients had healthcare related BSI (lines 139-144) since patients with BSI >48h had been recently discharged (if I understand correctly). If so, all patients should be regarded as having nosocomial infection and this variable should be reconsidered or discussed.
Answer: We clarified this issue by modifying the relative sentences stating that all patients had a secondary infection that occurred <=3 or >3 days after diagnosis of COVID-19. See page 3, lines 134 to 138 of the revised manuscript.
-A simple statistical analysis is missing in the sentence of lines 244-246/Figure 3a, and lines 248-250.
Answer: As suggested, we added statistical analysis in the two sentences. See page 8, line 256; and page 9, lines 289 to 291 of the revised manuscript.
-Line 347 refers to a second wave but this may be restricted to the author’s country, since other countries have experienced additional waves. Consider using another term.
Answer: As suggested, we used an alternative term to “second wave”. See page 11, line 409 of the revised manuscript.
-Consider addressing the high rate of MDR infections in the discussion and comment on the relationship with local susceptibility patterns, which may relate to empirical treatments and may not apply to other environments.
Answer: As suggested, we added a comment about this relevant issue. See page 10, lines 321 to 327 of the revised manuscript. Consistently, a new reference (ref. no. 31) was added, and both the sequence of cited references in the text and the reference list were modified accordingly.
Reviewer 2 Report
The manuscript deals with BSI in COVID-19 pts and risk factors for mortality. The topic is of great interest due to the dramatic epidemiological situation generated by a high burden of admissions for COVID-19.
Main concerns:
- inclusion and esclusion criteria should be clearly shown. In the text they are reported, but it is not easily readible.
- CA-BSI: were all healthcare risk factors ruled out?
- What does it mean "secondary BSI"? Secondary to COVID-19?
- BSI were 46, but in the text 58 episodes were studied. Please, clarify
- lines 202-206: a nested case control study was performed? It should be explained better and included in the methods
- I suggest to delete some tables and to show only the findings in the text. The tables could be presented as supplements
- Figure 3 is the most important part of the manuscript and deserves few additional words in the discussion
Author Response
Dear Sir/Madam,
I send you the revised version of the manuscript (jcm-1158044) “Risk factors for mortality in adult COVID-19 patients who develop bloodstream infections mostly caused by antimicrobial-resistant organisms: analysis at a large teaching hospital in Italy” by Posteraro et al. according to the reviewers' comments. All changes made in the manuscript were clearly highlighted, using the “Track Changes” function in Microsoft Word, so that they are easily visible to the editor and reviewers.
Below, you can find a short cover letter detailing any changes, for the benefit of the editor and reviewers. We detailed the revisions that have been made, citing the line number and exact change, so that the editor can check the changes expeditiously.
The manuscript deals with BSI in COVID-19 pts and risk factors for mortality. The topic is of great interest due to the dramatic epidemiological situation generated by a high burden of admissions for COVID-19.
Main concerns:
- Inclusion and exclusion criteria should be clearly shown. In the text, they are reported, but it is not easily readable.
Answer: As suggested, we specified the criteria to make them easily readable. See page 2, lines 77 to 80 of the revised manuscript.
- CA-BSI: were all healthcare risk factors ruled out?
Answer: All healthcare risk factors were ruled out to make that CA-BSI could be certainly defined. Accordingly, no CA-BSI was identified in our setting. To avoid confusion, the term “community-acquired” was deleted as appropriate. See page 3, lines 135 to 136.
- What does it mean "secondary BSI"? Secondary to COVID-19?
Answer: “Secondary BSI” does mean “infection secondary to SARS-CoV-2 infection (COVID-19)”. We clarified this point when necessary. See page 4, lines 167 to 168 and 170; and page 9, lines 298 to 299 of the revised manuscript.
- BSI were 46, but in the text, 58 episodes were studied. Please, clarify.
Answer: We specified that the patients with BSI were 46 but the BSI episodes studied in total were 58 because some patients had multiple BSI episodes. See page 6, line 190; and Figure 2 legend (page 7, line 198) of the revised manuscript.
- Lines 202-206: a nested case control study was performed? It should be explained better and included in the methods.
Answer: As suggested, we clarified this issue in the Materials and Methods section. See page 2, lines 80 to 83 of the revised manuscript.
- I suggest to delete some tables and to show only the findings in the text. The tables could be presented as supplements.
Answer: As suggested, we moved Tables 2 and 3 to Supplementary Material (Tables were renamed Table S2 and Table S4). Consequently, formerly named Table S2 was renamed Table S3.
- Figure 3 is the most important part of the manuscript and deserves few additional words in the discussion.
Answer: As suggested, we added some sentences to comment Figure 3. See page 10, lines 352 to 355 of the revised manuscript.